# Ex vivo anticoagulants affect human blood platelet biomechanics with implications for high-throughput functional mechanophenotyping

Laura Sachs[1], Jan Wesche[1], Lea Lenkeit[1], Andreas Greinacher [1], Markus Bender [2], Oliver Otto [3,4✉] & Raghavendra Palankar [1✉]

Inherited platelet disorders affecting the human platelet cytoskeleton result in increased bleeding risk. However, deciphering their impact on cytoskeleton-dependent intrinsic biomechanics of platelets remains challenging and represents an unmet need from a diagnostic and prognostic perspective. It is currently unclear whether ex vivo anticoagulants used during collection of peripheral blood impact the mechanophenotype of cellular components of blood. Using unbiased, high-throughput functional mechanophenotyping of single human platelets by real-time deformability cytometry, we found that ex vivo anticoagulants are a critical pre-analytical variable that differentially influences platelet deformation, their size, and functional response to agonists by altering the cytoskeleton. We applied our findings to characterize the functional mechanophenotype of platelets from a patient with Myosin Heavy Chain 9 (*MYH9*) related macrothrombocytopenia. Our data suggest that platelets from *MYH9* p.E1841K mutation in humans affecting platelet non-muscle myosin heavy chain IIa (NMMHC-IIA) are biomechanically less deformable in comparison to platelets from healthy individuals.

[1] Institute for Immunology and Transfusion Medicine, University Medicine Greifswald, Fleischmannstr.8, 17475 Greifswald, Germany. [2] Institute of Experimental Biomedicine - Chair I, University Hospital and Rudolf Virchow Center, Würzburg, Germany. [3] Zentrum für Innovationskompetenz – Humorale Immunreaktionen bei Kardiovaskulären Erkrankungen, Universität Greifswald, Fleischmannstr. 42, 17489 Greifswald, Germany. [4] Deutsches Zentrum für Herz-Kreislauf-Forschung e.V., Standort Greifswald, Universitätsmedizin Greifswald, Fleischmannstr. 42, 17489 Greifswald, Germany. ✉email: oliver.otto@uni-greifswald.de; raghavendra.palankar@med.uni-greifswald.de

Blood platelets are anuclear, discoidal multifunctional cellular fragments (1–3 μm in diameter) generated by bone marrow megakaryocytes and released into blood circulation[1]. On exposed extracellular matrix at the sites of the vascular breach, rapid recruitment of platelets is essential for forming a primary hemostatic plug. However, under pathological procoagulatory conditions, platelets contribute to intravascular thrombosis, a leading cause of cardiovascular complications and morbidities[2–4]. Platelets function as complex biological sensor and actuator units that respond to a broad spectrum of physicochemical stimuli via ligand-receptor-mediated interactions (i.e., outside-in signaling) and mechanotransduction events (i.e., both outside-in and inside-out signaling)[5–7]. This complex interplay results in the coordinated regulation of signaling kinetics, including cytoskeletal remodeling that initiates platelet adhesion, activation, spreading, and platelet contraction[8].

It has been well established that cytoskeleton-dependent biomechanics governs diverse aspects of platelet function during hemostasis and thrombosis[9,10]. Beyond this, the significance of platelet cytoskeletal integrity and its functional role in platelet-mediated innate immune responses such as mechano-scavenging, host defense during platelet-bacteria interactions, and vascular surveillance is emerging[11–13]. Recent studies have also demonstrated changes in platelet biomechanical properties, and subsequent defective mechanotransduction may serve as a biophysical marker for assessing bleeding risk in individuals with inherited platelet cytoskeletal defects[14]. Thus, deciphering cytoskeleton-dependent intrinsic biomechanical properties of platelets is highly desirable for broadening our understanding of the functional role of platelets in physiological and pathological processes and from translationally relevant diagnostic and prognostic perspectives[6,10].

Currently, a wide array of biophysical methods is available for the investigation of platelet biomechanics. They include micropipette aspiration[15–17], atomic force microscopy[18–20], scanning ion conductance microscopy[21,22], traction force microscopy[23,24], including flexible micropost arrays[25–27]. Although these methods have proven valuable in advancing our insights into platelet biomechanics, these are technically demanding, labor-intensive, and mostly limited to analysis of adherent platelets[28]. Besides, these methods also lack throughput, which results in implicit bias during single platelet measurements resulting from undersampling of innate heterogeneity found in donor platelet populations[29–31].

The recently introduced on-chip, high-throughput real-time fluorescence and deformability cytometry (RT-FDC) has rapidly emerged as a biophysical method to address these challenges[32,33]. RT-FDC enables continuous on-the-fly mechanophenotyping of single cells at real-time analysis rates exceeding 1000 cells/s combined with the capability of achieving molecular specificity through the application of fluorescent probes, which further opens up exciting possibilities[34–37].

However, on-chip deformability cytometry and other biophysical methods have not been well standardized regarding preanalytical variability in sample preparation of cells from peripheral blood, including platelets[38]. Specifically, it is unclear whether different ex vivo anticoagulants commonly used during blood sampling influence blood platelet biomechanics. Using high-throughput functional mechanophenotyping of single platelets in RT-FDC, here we demonstrate that ex vivo anticoagulants differentially impact intrinsic biomechanical properties (i.e., deformation and size) of human platelets. Besides this, we establish a link between platelet functional mechanophenotype, particularly their deformation and associated activation profiles as well as functional response in resting platelets and after activation with platelet agonist, respectively, in different ex vivo anticoagulants.

We explain these findings by showing that ex vivo anticoagulants and platelet activation alter the content and subcellular organization of major platelet cytoskeletal components such as actin cytoskeleton and marginal band tubulin ring. Furthermore, in a potentially diagnostically relevant development, using MYH9 related macrothrombocytopenia as a model for an inherited human platelet cytoskeletal disorder, affecting platelet non-muscle myosin heavy chain IIa (NMMHC-IIA), we demonstrate that the choice of ex vivo anticoagulant may strongly impact the outcomes of mechanophenotyping.

## Results

**Ex vivo anticoagulants affect human platelet deformation and size.** We first evaluated the effects of ex vivo anticoagulants on platelet deformation, and their corresponding size in live non-stimulated (i.e., resting) platelets in PRP by RT-FDC prepared from blood collected in ACD-A, Na-Citrate, $K_2$-EDTA, Li-Heparin, and r-Hirudin (Fig. 1a). Non-stimulated platelets showed deformation of 0.127 ± 0.033 (mean ± SD, $n = 6$ donors) in ACD-A, 0.111 ± 0.025 in Na-Citrate, and 0.1 ± 0.023 in r-Hirudin, which was higher in comparison to the lower deformation of platelets at 0.071 ± 0.016 observed in Li-Heparin and 0.037 ± 0.01 in $K_2$-EDTA. (Fig. 1b and Supplementary Fig. 4 for statistical distribution plots of individual donors). Assessment of corresponding platelet size in different ex vivo anticoagulants from non-stimulated platelets revealed differences in platelet size measuring at 5.035 ± 0.49 μm² (mean ± SD, $n = 6$ donors) in ACD-A in comparison to $K_2$-EDTA and Li-Heparin where platelets were smaller in size measuring at 4.158 ± 0.241 and 4.337 ± 0.344 μm², respectively (Fig. 1c and Supplementary Fig. 5 for statistical distribution plots of individual donors).

Next, to test whether agonist-induced platelet activation leads to measurable changes in platelet deformation and their corresponding size depending on the type of ex vivo anticoagulant, TRAP-6 was used. Platelet activation by TRAP-6 resulted in a noticeable decrease in platelet deformation and a concomitant reduction in platelet size in all ex vivo anticoagulants except for $K_2$-EDTA (Fig. 1d–f). Assessment of fold change in platelet deformation before and after TRAP-6 stimulation showed a decrease in platelet deformation by a factor of 2.76 ± 0.64 (mean ± SD, $n = 6$ donors) in ACD-A, 2.58 ± 0.49 in Na-Citrate, 1.72 ± 0.47 Li-Heparin, and 2.27 ± 0.45 in r-Hirudin (Fig. 1g). On the contrary, in $K_2$-EDTA, TRAP-6 stimulation resulted in a minimal fold change in platelet deformation by a factor of 1.14 ±0.33 (Fig. 1g). Similarly, platelet size decreased upon TRAP-6 stimulation by a factor of 1.28 ± 0.13 (mean ± SD, $n = 6$ donors) in ACD-A, 1.14 ± 0.14 in Na-Citrate and 1.18 ± 0.16 in r-Hirudin, while it remained unchanged at 0.98 ± 0.08 in $K_2$-EDTA and 1.04 ± 0.04 in Heparin (Fig. 1h). Furthermore, the changes in platelet shape observed in the RT-FDC differed between the ex vivo anticoagulants in non-stimulated and TRAP-6 stimulated platelets (Supplementary Fig. 6 representative bright-field images of single platelets in measurement channel overlaid with contour).

**Platelet deformation in response to platelet activation.** In non-stimulated platelets, basal CD62P surface expression was not altered between all ex vivo anticoagulants even though platelets in $K_2$-EDTA exhibited decreased deformation relative to other ex vivo anticoagulants (Fig. 2a–c). Upon activation of platelets by TRAP-6, a decrease in platelet deformation with a concomitant increase in CD62P surface expression and CD62P % positive platelets was observed in all ex vivo anticoagulants except in $K_2$-EDTA (Fig. 2d–f). Assessment of fold change in CD62P expression levels showed an increase by a factor of 18.19 ± 8.88 (mean ± SD, $n = 6$ donors) in ACD-A, 21.48 ± 8.54 in Na-Citrate, 9.82 ± 7.78 in Li-Heparin, and

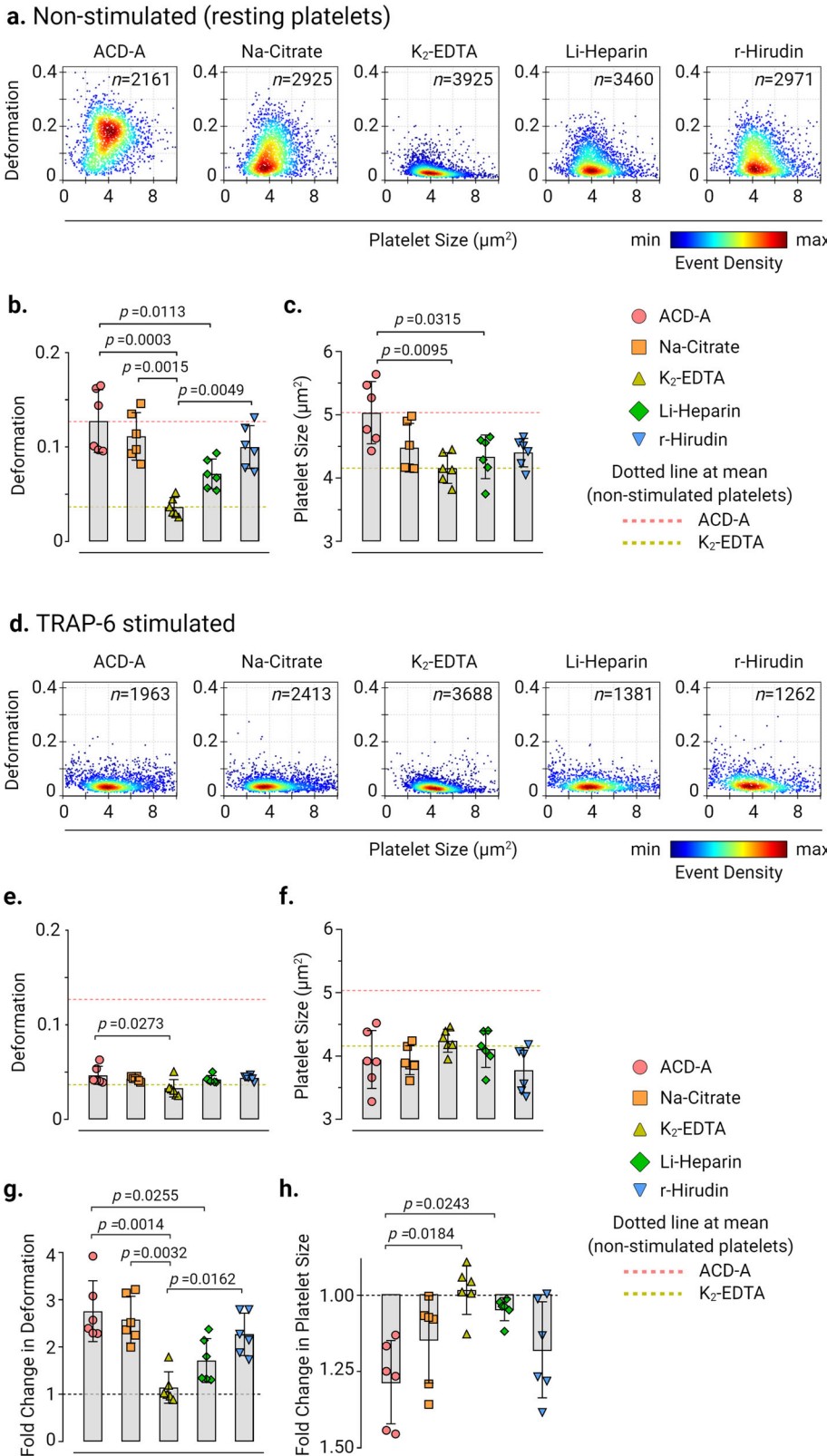

**Fig. 1 Ex vivo anticoagulants influence the deformation and size of human platelets.** Representative KDE scatter plots of deformation and size of live single platelets in PRP prepared from whole blood collected in different ex vivo anticoagulants from **a** non-stimulated (i.e., resting) platelets and **d** after stimulation with TRAP-6 (n = number of single platelets from the same donor measured for each condition). Color coding of event density in scatter plots indicates a linear density scale from min (blue) to max (dark red). Summary data points show the median values of individual donors, and bar plots show mean ± SD of platelet deformation and size from non-stimulated (**b**) and (**c**) and TRAP-6 stimulated platelets (**e**) and (**f**), respectively (n = 6 donors). Fold change in platelet deformation and size upon stimulation with TRAP-6 are shown in (**g**) and (**h**), respectively, where dotted control baseline = 1 (n = 6 donors). Statistical analysis: mixed-effects model (restricted maximum likelihood, REML) followed by Tukey's multiple comparisons tests, with single pooled variance and p < 0.05 was considered significant.

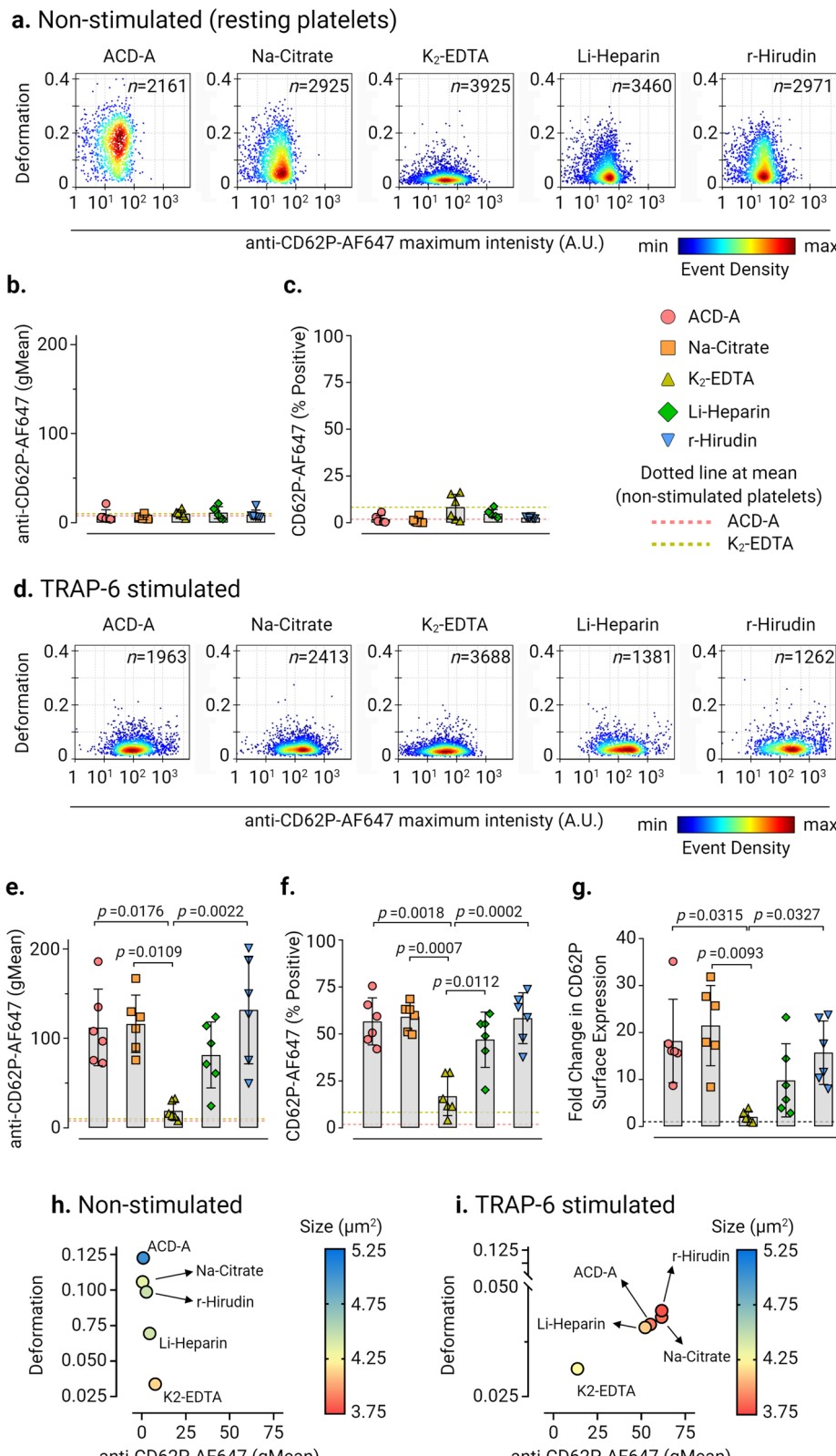

15.72 ± 6.76 in r-Hirudin in comparison to a fold change of 2.03 ± 1.16 in $K_2$-EDTA (Fig. 2g). Multivariate analysis of continuous variables from RT-FDC data from non-stimulated and TRAP-6 stimulated platelets (Fig. 2h, i) further confirmed platelet deformation, size, and CD62P expression is strongly affected in $K_2$-EDTA.

Next, we assessed conformational changes in platelet integrin $\alpha_{IIb}\beta_3$ as a marker for platelet activation by PAC-1 antibody

binding (Fig. 3). Baseline activation levels of integrin $\alpha_{IIb}\beta_3$ in non-stimulated platelets (Fig. 3a–c) were highest in Li-Heparin (PAC-1-FITC gMean of 82.48 ± 12.16 and PAC-1-FITC % positive platelets at 36.02 ± 9.3%, mean ± SD, $n = 6$ donors). In contrast, $K_2$-EDTA showed the lowest basal activation of integrin $\alpha_{IIb}\beta_3$ of all ex vivo anticoagulants. In TRAP-6 stimulated platelets, PAC-1-FITC binding and PAC-1-FITC % positive

**Fig. 2 Platelet deformation and the corresponding CD62P surface expression upon activation in different ex vivo anticoagulants.** Representative KDE scatter plots of platelet deformation and CD62P expression on single platelets expression (plotted on log10 scale of maximum intensity in arbitrary units (A.U.) of anti-CD62P-AlexaFluor647 antibody) in PRP different ex vivo anticoagulants in **a** non-stimulated (i.e., resting) platelets and **d** upon TRAP-6 stimulation ($n$ = number of single platelets from the same donor measured for each condition). Color coding of event density in scatter plots indicates a linear density scale from min (blue) to max (dark red). Summary graphs show of median values of individual donors, while bar graphs show mean ± SD of geometric mean fluorescence intensity (gMean) of CD62P expressing platelets (**b**) and (**e**) and percent positive platelets (**c**) and (**f**) above the cut-off of 5000 events or 10 min in non-stimulated and TRAP-6 stimulated platelets, respectively ($n$ = 6 donors). Fold change in CD62P surface expression on platelets upon stimulation with TRAP-6 are shown in (**g**) where dotted control baseline = 1 ($n$ = 6 donors). Multivariate analysis plots of continuous variables from RT-FDC, (**h**) and (**i**) of non-stimulated and TRAP-6 stimulated platelets, respectively, displaying the relationships between platelet deformation, size, and related CD62P surface expression levels in different ex vivo anticoagulants. (Data represents median values of individual variables from $n$ = 6 donors). Statistical analysis: mixed-effects model (restricted maximum likelihood, REML) followed by Tukey's multiple comparisons tests, with single pooled variance and $p < 0.05$ was considered significant.

platelets increased significantly ($p < 0.0001$) in all ex vivo anticoagulants in comparison to $K_2$-EDTA (Fig. 3d–f). Furthermore, fold change in PAC-1 binding to platelets after TRAP-6 stimulation increased by a factor of 2.86 ± 0.82 (mean ± SD, $n$ = 6 donors) in ACD-A, 3.39 ± 0.9 in Na-Citrate, 2.64 ± 0.7 in Li-Heparin, and 3.55 ± 0.98 in r-Hirudin in comparison to non-stimulated platelets, but did not increase in $K_2$-EDTA (Fig. 3g). Also, multivariate analysis of continuous variables from RT-FDC data from non-stimulated and TRAP-6 stimulated platelets (Fig. 3h, i) revealed together with platelet deformation, size, and PAC-1 binding is strongly affected in $K_2$-EDTA. Our results concerning the reduced binding of PAC-1 antibody to platelet integrin αIIbβ3 are consistent with previous observations in potent chelators of divalent cations such as EDTA[39].

**Decreased platelet deformation is an indicator of alterations in platelet cytoskeletal organizations and F-actin content.** Fluorescence CLSM imaging and subsequent line profile analysis of non-stimulated platelets in ACD-A, Na-Citrate, Li-Heparin, and r-Hirudin showed discoidal morphology, a uniform intracellular distribution of F-actin (Phalloidin, green), and a well-defined subcortical marginal band microtubule ring (α-tubulin, edge-to-edge fluorescence intensity signal in magenta) (Fig. 4a and Supplementary Fig. 8). In contrast, platelets in $K_2$-EDTA platelets lost their discoidal shape and were comparatively smaller. Besides, we observed a significant increase ($p < 0.001$) in subcortical localization of F-actin (Fig. 4b and Supplementary Fig. 8) and coiling of microtubule ring indicated by white arrowhead in grayscale subfigures in Fig. 4a and edge-to-edge fluorescence line profile intensity in Fig. 4c and Supplementary Fig. 8 for $K_2$-EDTA. Upon TRAP-6 stimulation, platelets showed substantial morphological changes compared to their non-stimulated counterparts (Fig. 4d). Line profile analysis of fluorescence intensities further revealed a significantly increased F-actin localization ($p < 0.001$) at subcortical regions and a decrease in edge-to-edge coiling of microtubule ring in all ex vivo anticoagulants except for $K_2$-EDTA (Fig. 4e, f and Supplementary Fig. 8).

Next, we analyzed the total F-actin content in non-stimulated and TRAP-6 stimulated platelets in different ex vivo anticoagulants by flow cytometry (Fig. 5a, b and Supplementary Fig. 7). We observed a significantly higher F-actin content in non-stimulated platelets in $K_2$-EDTA (Phalloidin AF647 fluorescence gMean: 162.2 ± 30.47, mean ± CD from $n$ = 6 donors) in comparison to non-stimulated platelets measuring at 110.7 ± 32.11, $p$ = 0.036, in ACD-A, 107 ± 23.86, $p$ = 0.0285 in Na-Citrate and 94.24 ± 28.91, $p$ = 0.0023 in r-Hirudin (Fig. 5a). The basal F-actin content of platelets in Li-Heparin was found to be relatively higher (Phalloidin AF647 fluorescence gMean: 146.5 ± 9, mean ± CD from $n$ = 6 donors) than in ACD-A and Na-Citrate, statistically significant differences ($p$ = 0.024) were apparent between Li-Heparin and r-Hirudin in non-stimulated

platelets (Fig. 5a). TRAP-6 stimulation resulted in the increase of total F-actin in platelets by a factor of 2.18 ± 0.28 in ACD-A, 2.13 ± 0.38 in Na-Citrate, and 2.1 ± 0.27 in r-Hirudin. In contrast, only a minor change in total F-actin content by a factor of 1.49 ± 0.26 in Li-Heparin and 0.96 ±0.14 in $K_2$-EDTA was observed (Fig. 5b, c and Supplementary Fig. 7).

Next, a multivariate analysis of continuous variables was performed to verify whether the changes in actin polymerization status, i.e., total F-actin content measured by flow cytometry, reflect the observed differences in platelet deformation and their corresponding size by RT-FDC in different ex vivo anticoagulants. As shown, non-stimulated (i.e., resting) platelets (Fig. 5d) were found to deform more with low basal F-actin content in ACD-A, Na-Citrate r-Hirudin than those in Li-Heparin. Furthermore, platelets in $K_2$-EDTA deformed least with higher basal F-actin content and smallest in size. Under TRAP-6 stimulation, except in $K_2$-EDTA, platelets in all ex vivo anticoagulants showed decreased deformation, a smaller size, and increased total F-actin content (Fig. 5e).

**Actin disassembly by LatB increases platelet deformation.** Next, we used latrunculin B (LatB) an actin polymerization inhibitor, to gain mechanistic insights into whether impairing actin polymerization influences biomechanical deformation and functional response of platelets in different ex vivo anticoagulants (Fig. 6a–d and Supplementary Fig. 9). In non-stimulated platelets, LatB significantly increased ($p < 0.001$) platelet deformation in all ex vivo anticoagulants compared to their respective vehicle control. Similarly, in contrast to the vehicle control (6e) in the presence of LatB, TRAP-6 stimulated platelets deformed more (Fig. 6g). Furthermore, only in ACD-A, LatB induced increase in platelet deformation was higher in resting platelets in all donors (Fig. 6f) compared to all ex vivo anticoagulants. In addition, in the presence of LatB basal CD62P surface expression and PAC-1 binding were unaltered in all ex vivo anticoagulants (Supplementary Fig. 9a, c). Although after TRAP-6 stimulation, LatB treated platelets in all donors showed a markedly increased deformation, however, only in $K_2$-EDTA platelets deformation was significant ($p$ = 0.0066) compared to other ex vivo anticoagulants (Fig. 6h). Furthermore, TRAP-6 induced functional response of LatB treated platelets showed reduced CD62P surface expression in all ex vivo anticoagulants compared to respective vehicle controls but remained unchanged in $K_2$-EDTA (Supplementary Fig. 9b).

**ACD-A, but not $K_2$-EDTA, allows mechanophenotyping of MYH9 related disease mutations in human platelets.** By RT-FDC, we next analyzed platelets from an individual with MYH9 p.E1841K mutation in the rod region of NMMHC-IIA, an essential platelet cytoskeletal protein[40]. In ACD-A, MYH9 p.E1841K platelets

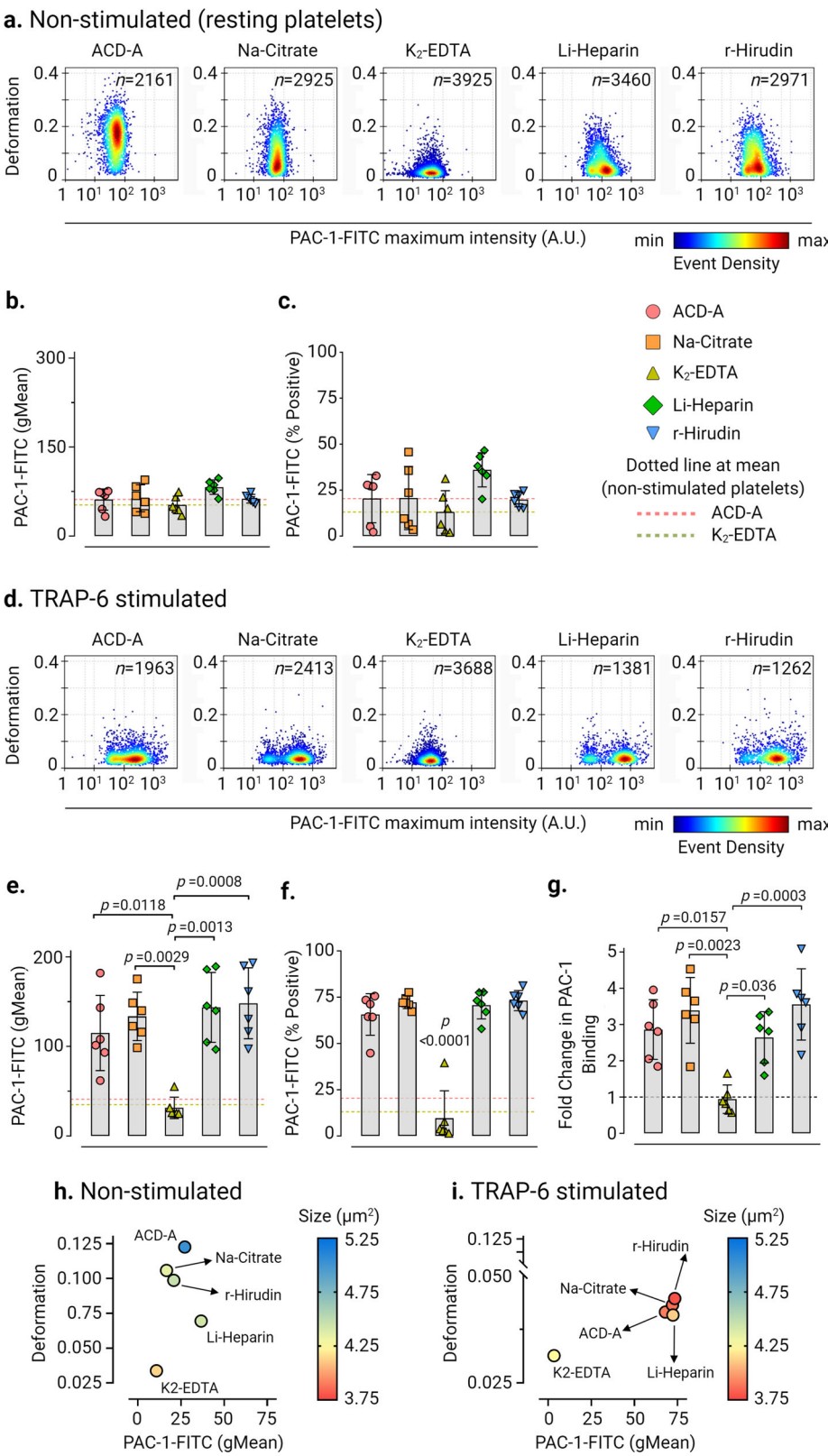

in comparison to platelets from healthy controls deform less (0.068, median $n = 955$ single platelets vs. 0.122, median $n = 2326$ single platelets) and are larger (5.77 μm², median $n = 955$ single platelets vs. 4.05 μm², median $n = 2326$ single platelets) (Fig. 7a and Supplementary Fig. 10a) under non-stimulated (i.e., resting) conditions. With TRAP-6 stimulation, platelets from the individual with *MYH9* p.E1841K in ACD-A deform further less (0.036, median $n = 1112$

single platelets), but intriguingly their size increased (6.585 μm², median $n = 1112$ single platelets). In contrast, the platelets from the healthy individual showed decreased deformation (0.0455, median $n = 720$ single platelets) and size (2.595 μm²) (Fig. 7b and Supplementary Fig. 10b). On the other hand, in $K_2$-EDTA, non-stimulated healthy control platelets showed ≈ 3-fold decreased deformation (0.036, median $n = 1960$ single platelets) and while *MYH9* p.E1841K

**Fig. 3 Platelet deformation and activation-induced exposure of the conformational epitope of the integrin αIIbβ3 is strongly influenced by ex vivo anticoagulants.** Representative KDE scatter plots of platelet deformation and their corresponding activation status as a readout for binding of integrin αIIbβ3 specific ligand-mimetic PAC-1 antibody (plotted on log10 scale of maximum intensity in arbitrary units (AU) of PAC-1-FITC antibody) on single platelets in different ex vivo anticoagulants in (**a**) non-stimulated (i.e., resting) platelets and upon stimulation TRAP-6 (**d**) from a single donor (n= number of single platelets from the same donor measured for each condition). Color coding of event density in scatter plots indicates a linear density scale from min (blue) to max (dark red). Summary graphs show of median values of individual donors, while bar graphs show mean ± SD of geometric mean fluorescence intensity (gMean) of PAC-1-FITC antibody bound to platelets (**b**) and (**e**) and PAC-1-FITC antibody percent positive platelets (**c**) and (**f**) above the cut-off of 5000 events or 10 min in non-stimulated and TRAP-6 stimulated platelets, respectively ($n = 6$ donors). Fold change in PAC-1 Binding on platelets upon stimulation with TRAP-6 from six donors are shown in (**g**) where dotted control baseline = 1 ($n = 6$ donors). Multivariate analysis plots of continuous variables from RT-FDC, (**h**) and (**i**) of non-stimulated and TRAP-6 stimulated platelets, respectively, displaying the relationships between platelet deformability, size, and PAC-1 antibody biding to integrin αIIbβ3 in different ex vivo anticoagulants. (Data represents median values of individual variables from $n = 6$ donors). Statistical analysis: mixed-effects model (restricted maximum likelihood, REML) followed by Tukey's multiple comparisons tests, with single pooled variance and $p < 0.05$ was considered significant.

platelets showed a ≥ 3-fold decreased deformation (0.0195, median $n = 2406$ single platelets) in comparison to their counterparts in ACD-A (Fig. 7c, e and Supplementary Fig. 11a).

Besides, in $K_2$-EDTA, platelets from the healthy individual became smaller (3.27 μm², median $n = 1960$ single platelets). In contrast, platelets from the *MYH9* p.E1841K individual were slightly increased in their size (7.515 μm², median $n = 2406$ single platelets) compared to non-stimulated platelets in ACD-A. Furthermore, TRAP-6 stimulation of platelets in $K_2$-EDTA only resulted in minor changes in platelet deformation and size compared to their non-stimulated counterparts (Fig. 7d and Supplementary Fig. 11b). Although differences in platelet deformation and size were apparent between the platelets from *MYH9* p.E1841K patient and healthy control in ACD-A; their CD62P surface expression levels and PAC-1 binding in response to TRAP-6 were comparable (Supplementary Fig. 12 and Fig. 13).

Consistent with our observations reported above, in $K_2$-EDTA platelets from both *MYH9* p.E1841K patient and healthy control failed to respond to TRAP-6 (Supplementary Figs. 12 and 13). Furthermore, in ACD-A, assessment of F-actin content revealed a higher basal total F-actin content in non-stimulated platelets from *MYH9* p.E1841K patient at 192.86 (Phalloidin AF647 gMean) compared to the healthy control at 154.96, which upon TRAP-6 stimulation increased to 250.16 and 286.36, (Supplementary Fig. 14a). On the other hand, in $K_2$-EDTA, the basal total F-actin content in non-stimulated platelets from *MYH9* p.E1841K patient was found to be at 250.42 (Phalloidin AF647 gMean) and for healthy control at 206.6, that remained unchanged upon TRAP-6 stimulation (Supplementary Fig. 14b).

## Discussion

Clinically relevant translational applications for the diagnosis of disease states by morpho-rheological and biomechanical characterization of a variety of cells by RT-DC are fast emerging. These include biomechanical differentiation for diagnosis of hereditary spherocytosis and malaria infections in peripheral blood cells[41] and in biophysical fingerprinting of primary human skeletal stem cells from bone marrow[42]. In addition, the feasibility of RT-DC in routine quality control of platelet concentrates and transplantable hematopoietic stem cells in blood banks is currently being tested[43]. More recently Kubankova et al. using RT-DC performed biomechanical fingerprinting of erythrocytes, lymphocytes, monocytes, neutrophils, and eosinophils in clinical induced by severe acute respiratory syndrome corona virus2 (COVID-19 disease)[44]. The study results showed highly deformable lymphocytes and neutrophils, increased size of monocytes, lymphocytes, and neutrophils, and the appearance of smaller and less deformable erythrocytes.

The present study shows that $K_2$-EDTA and Li-Heparin should not be used as ex vivo anticoagulants for studies on human platelet biomechanical properties. Platelets collected in ACD-A, Na-Citrate, or r-Hirudin may be used for biomechanical studies. Still, due to minor differences in the effects on platelets, results cannot be directly compared between platelets anticoagulated with these different anticoagulants. Platelets anticoagulated with Li-Heparin show some differences in their biomechanical characteristics compared to platelets in ACD-A, Na-Citrate, or r-Hirudin. Heparinized platelets show a two-fold higher F-actin content, decreased deformation, and higher PAC-1 expression. The most crucial difference between Li-Heparin and r-Hirudin compared to the other anticoagulants is that Li-Heparin and r-Hirudin do not chelate calcium. However, the apparent discrepancies between Li-Heparin and r-Hirudin indicate that Li-Heparin is inducing artifacts in the biomechanical properties of platelets. One explanation is the strong negative charge of heparin and its binding to αIIbβ3, which triggers αIIbβ3-mediated outside-in signals and thus initiates cytoskeletal reorganization[45,46]. Platelets collected in $K_2$-EDTA have the highest F-actin content under resting conditions in comparison to the other anticoagulants and show the lowest deformation. Our observations are in agreement with previous studies, which have demonstrated $K_2$-EDTA induced ultrastructural changes of the surface-bound canal system (narrowing and dilatation of the OCS) and an irreversible dissociation of the αIIbβ3 complexes[47–50]. It is possible that the high content of F-actin in non-stimulated and TRAP-6 stimulated platelets and the associated platelet deformation could be explained by the irreversible dissociation of the αIIbβ3 complex and the associated cytoskeletal reorganization.

To support these findings, we employed LatB, an inhibitor of actin polymerization, to provide biomechanical insights into whether impairing actin polymerization influences the biomechanical deformation and functional response of platelets in different ex vivo anticoagulants, thus providing a direct link between platelet deformability and F-actin content. Our results show that LatB increased platelet deformability in unstimulated platelets in all ex vivo anticoagulants compared with the respective vehicle control. Similarly, TRAP-6-stimulated platelets were more deformable in the presence of LatB than in the vehicle control. We also investigated the effect of actin depolymerization by LatB on the surface expression of CD62P and integrin activation by PAC-1 binding. CD62P surface expression after TRAP-6 stimulation was reduced by LatB treatment in the anticoagulants ACD-A, Li-Heparin, and r-Hirudin. In $K_2$-EDTA, no change in CD62P expression occurred after TRAP-6 stimulation in the vehicle control and LatB, confirming our data that CD62P expression is impaired after TRAP-6 activation in $K_2$-EDTA. Our results are consistent with the data of Woronowicz et al. showing that LatA (an isomer of latrunculin that binds gelsolin) inhibits alpha-granule secretion by disrupting the actin cytoskeleton[51]. We also observed a marked reduction in PAC-1 binding after LatB treatment of resting platelets with the anticoagulants

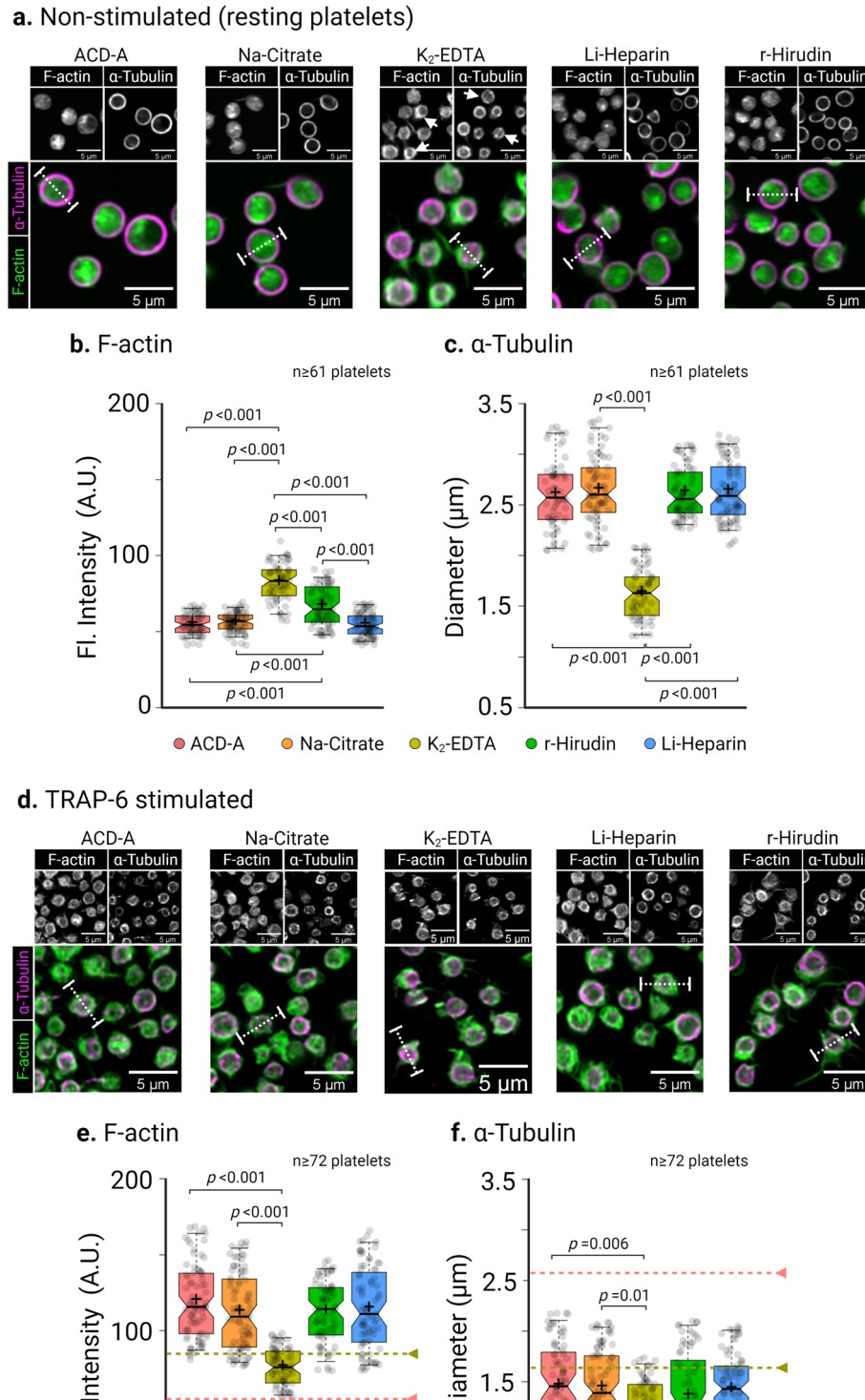

**Fig. 4 Cytoskeletal organization in resting and TRAP-6 stimulated human platelets are altered in different ex vivo anticoagulants.** Representative confocal laser scanning fluorescence microscopic images of F-actin (green) distribution and marginal band α-tubulin (magenta) organization of human platelets in different ex vivo anticoagulants in (**a**) non-stimulated (resting platelets) and (**d**) 10 min after TRAP-6 stimulation. Line profile fluorescence intensity distribution analysis of F-actin and edge-to-edge distance of α-tubulin (**b**) and (**c**) in non-stimulated platelets and after TRAP-6 stimulation (**e**) and (**f**). Dotted horizontal lines in (**e**) and (**f**) correspond to the median fluorescence intensity and diameter of respective parameters from non-stimulated platelets. Notch in the box plot and the plus sign depicts median and mean, respectively, and the interquartile ranges. Staggered dots in gray show the distribution of data points. Statistical assessment by Kruskal-Wallis-Test followed by Dunn's multiple comparisons test $p < 0.05$ was considered significant.

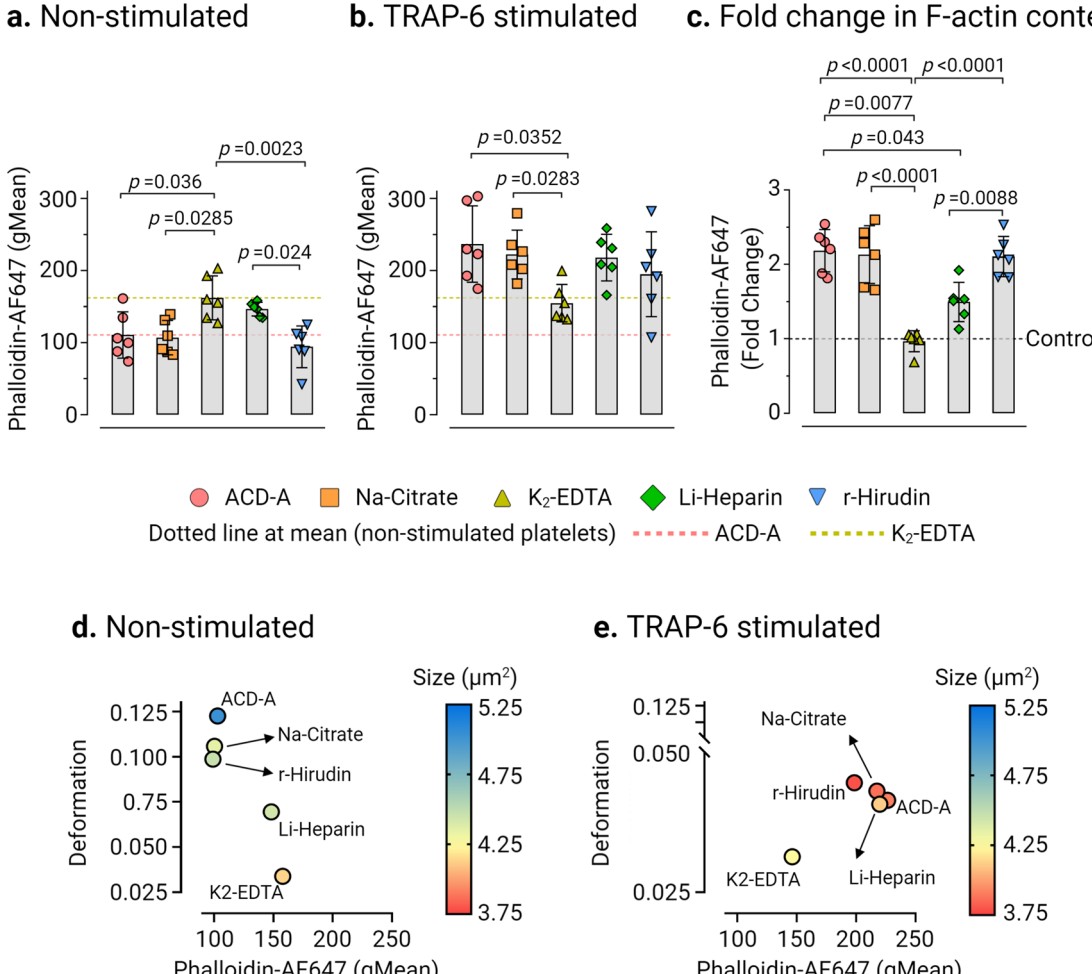

**Fig. 5 The change of F-actin content in platelets is an indicator of platelet deformation.** Comparison of platelet F-actin (Phalloidin AF647 fluorescence, gMean) measured by flow cytometry in **a** non-stimulated and **b** TRAP-6 stimulated platelets in different ex vivo anticoagulants. Fold change in phalloidin binding (**c**) after stimulation with TRAP-6, where control baseline = 1 and plots show mean ± SD from $n = 6$ donors. Multivariate analysis plots of continuous variables from RT-FDC and flow cytometry, (**d**) and (**e**) of non-stimulated and TRAP-6 stimulated platelets, respectively, displaying the relationships between platelet deformation, size, and the F-actin content in different ex vivo anticoagulants, shown as a measure of phalloidin binding. (Data represents median values of individual variables from $n = 6$ donors). Statistical assessment was performed by applying the mixed-effects model (restricted maximum likelihood, REML) followed by Tukey's multiple comparisons test, with single pooled variance and $p < 0.05$ was considered significant.

ACD-A, Li-Heparin, and r-Hirudin. Interestingly, an apparent decrease in PAC-1 binding is observed after TRAP-6 activation only with Li-heparin, confirming our hypothesis that Li-heparin induces cytoskeletal reorganization by binding to αIIbβ3, making platelets deform more.The difference in deformation of activated platelets between vehicle control and LatB in $K_2$-EDTA indicates that the increased F-actin content in resting platelets in $K_2$-EDTA is responsible for the increased deformation when actin polymerization is blocked by LatB, resulting in destabilization of the actin cytoskeleton.

The practical relevance of our findings is exemplified by the results obtained with *MYH9* p.E1841K platelets in ACD-A compared to $K_2$-EDTA. The non-stimulated platelets in ACD-A deform more than those in $K_2$-EDTA, even after TRAP-6 induced activation. We conclude that the anticoagulants $K_2$-EDTA and Li-Heparin are not suitable for the study of the human platelet cytoskeleton, while ACD-A, Na-Citrate, or r-Hirudin can be used. Concerning our reasoning behind choosing ACD-A over Na-Citrate as an ex vivo anticoagulant as a comparator to $K_2$-EDTA for the analysis of platelets from a patient with *MYH9* mutation in RT-FDC, there are several lines of experimental evidence which indicate ACD-A is superior to Na-Citrate in terms of

maintaining platelet physiology. In practice, blood collection systems anticoagulated with Na-Citrate are primarily used for studies of plasmatic coagulation and aggregation studies. Intriguingly, Na-Citrate may induce the formation of micro-aggregates, thus leading to a decrease in platelet count over time[52,53]. On the contrary, ACD-A is a more physiological anticoagulant capable of maintaining platelet physiology and signal transduction mechanisms with minimal impact on platelet responsiveness to agonists[54]. These results may facilitate a comparison between different laboratories using shear-based deformability cytometry such as RT-FDC to address fundamental questions of platelet physiology and its relationship with biomechanical phenotype and may help to avoid artifacts when these new technologies are applied to investigate patients with platelet disorders.

## Conclusions
In summary, we can conclude that $K_2$-EDTA and Li-Heparin influence the biomechanics of platelets by decreasing the deformation and increasing actin polymerization of non-stimulated human platelets. It is recommended for the examination of the human platelet cytoskeleton to select an ex vivo anticoagulant

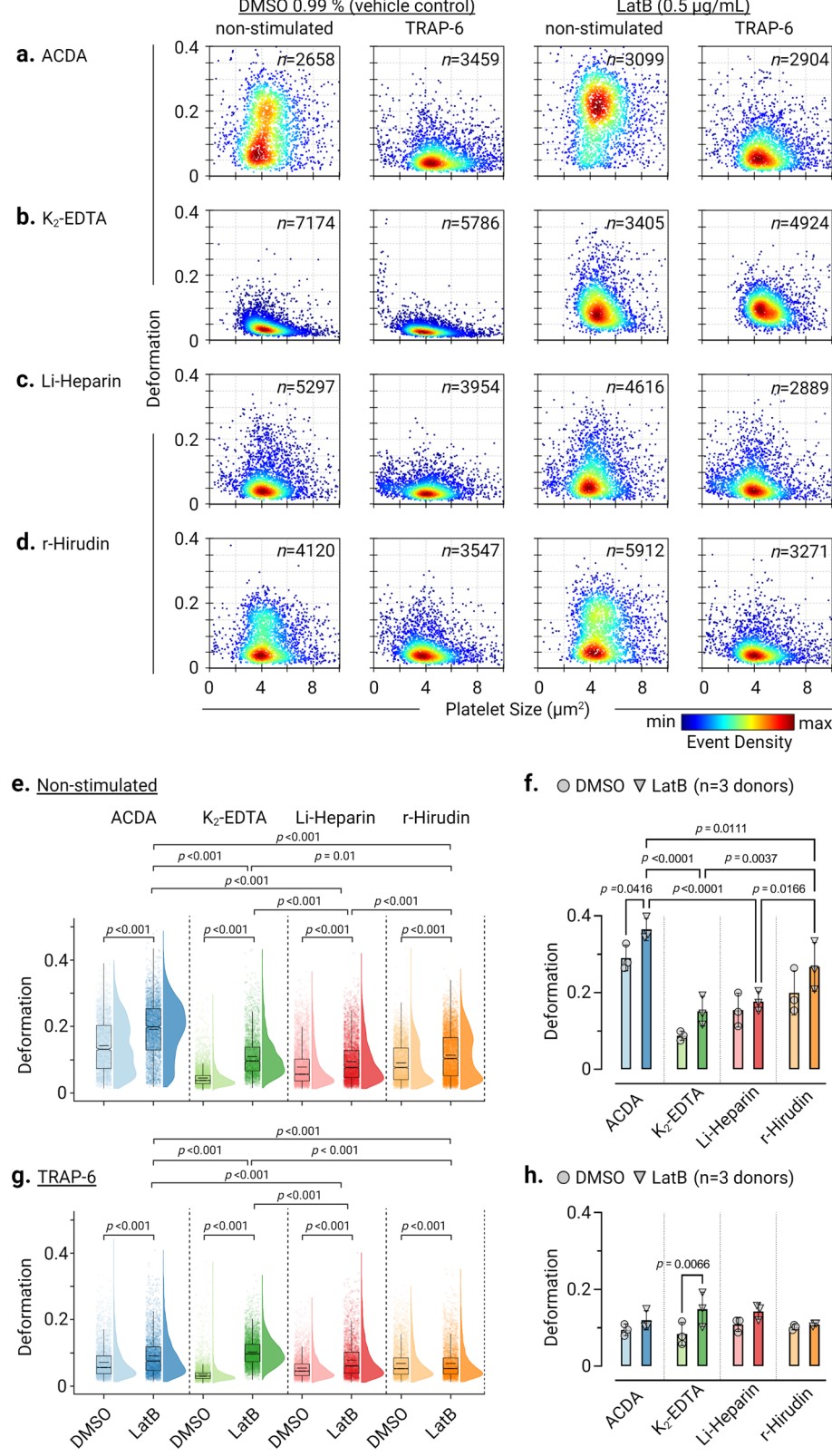

**Fig. 6 Inhibition of actin polymerization by LatB strongly affects platelet deformation.** KDE scatter plots showing the impact of LatB on single platelet deformation and size on non-stimulated (resting) platelets and after TRAP-6 stimulation compared to the vehicle control (DMSO 0.99 %) in **a** ACD-A, **b** $K_2$-EDTA, **c** Li-Heparin and in **d** r-Hirudin (n= number of single platelets). Statistical distribution plots for platelet deformation from non-stimulated platelets (**e**) and (**f**) after TRAP-6 stimulation (**g**) and (**h**). Notch in the box plot and the horizontal line depicts median and mean, respectively, and the interquartile ranges. The full distribution of the data for each parameter is depicted by half-violin plots and staggered dots. (Individual data points in (**f**) and (**h**) show median values of platelet deformation from $n = 3$ donors). Statistical assessment by Kruskal-Wallis-Test followed by Dunn's multiple comparisons test and repeated measures one-way ANOVA followed by Tukey's multiple comparisons test. $p < 0.05$ was considered significant.

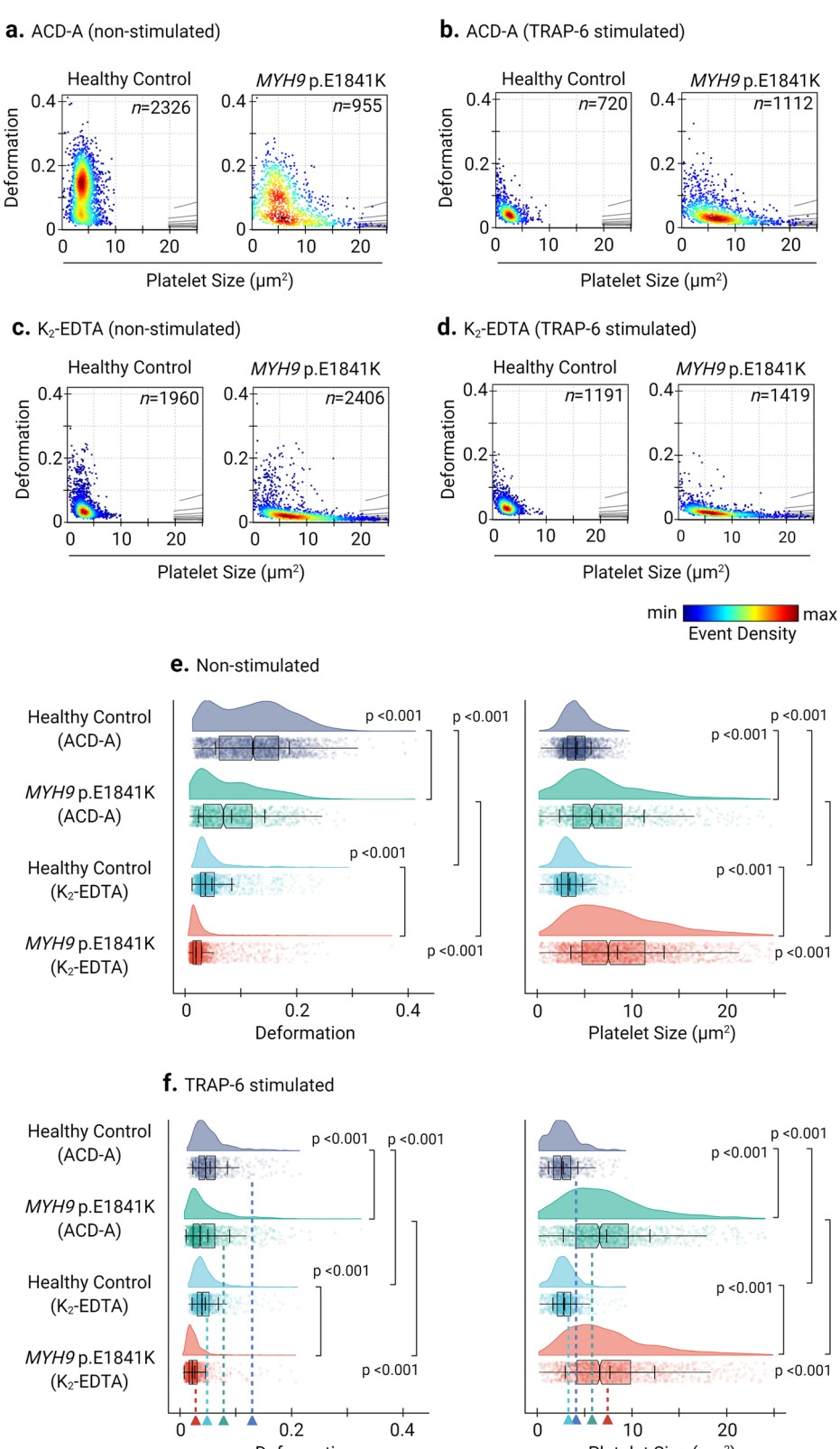

**Fig. 7 Deformability and size of platelets from a healthy individual and from a patient carrying *MYH9* p.E1841K mutation.** KDE scatter plots from RT-FDC measurements performed on the same day of displaying the distribution of single platelet deformability and their corresponding size between single platelets from a healthy individual (control) and from a patient carrying the mutation *MYH9* p.E1841K for non-muscle myosin heavy chain IIa, in ex vivo anticoagulant (**a**) ACD-A and (**c**) in $K_2$-EDTA before and after stimulation with platelet agonist TRAP-6, (**b**) and (**d**), respectively (*n* = number of single platelets). Statistical distribution plots for platelet deformation and size for (**e**) non-stimulated and (**f**) TRAP-6 stimulated platelets from each experimental condition from figure panels (**a**) through (**d**). Notch in the box plot and the horizontal line depicts median and mean, respectively, and the interquartile ranges. The full distribution of the data for each parameter is depicted by half-violin plots and staggered dots and dotted lines with arrowheads represent the median values of respective parameters from the non-stimulated platelets. Statistical assessment by Kruskal–Wallis test followed by Dunn's multiple comparisons test. *p* < 0.05 was considered significant.

such as ACD-A, Na-Citrate, or r-Hirudin and not to exchange it if possible since comparability of the results cannot be guaranteed. With the RT-FDC, we have a highly promising method to examine the platelet cytoskeleton in PRP, which according to our study, provides very solid and fast results.

## Methods

**Ethics**. The use of platelet-rich plasma (PRP) from healthy adult individuals and *MYH9* patients was approved by the ethics committee of the University Medicine Greifswald, Germany. All participants gave written, informed consent

**Platelet preparation**. The donors had not taken any medication in the previous ten days before blood collection. Whole blood was collected by venipuncture in BD Vacutainer® Tubes containing acid citrate dextrose solution A (ACD-A), 3.8% buffered trisodium citrate (Na-Citrate), 102 I.U. Lithium-Heparin (Li-Heparin), 1.8 mg/mL dipotassium ethylenediaminetetraacetic acid ($K_2$-EDTA), or 171 ATU/mL recombinant hirudin (r-Hirudin) (REVASC, Canyon Pharmaceuticals, USA). Whole blood was stored at room temperature for 15 min (at 45° angle to the horizontal surface) and then centrifuged ($120 \times g$ for 20 min at room temperature). PRP was transferred to a new polypropylene tube and incubated for 15 min at 37 °C. All experimental measurements were performed within 3 h of drawing the blood.

**Real-time fluorescence deformability cytometry (RT-FDC)**. The RT-FDC setup (AcCellerator, Zellmechanik Dresden, Germany) is built around an inverted microscope (Axio Observer A1, Carl Zeiss AG, Germany) mounted with a Zeiss A-Plan 100× NA 0.8 objective. The RT-FDC fluorescence module (Supplementary Fig. 1a) is equipped with 488 nm, 561 nm, 640 nm excitation lasers, and emission is collected at the following wavelengths: 500–550, 570–616, 663–737 nm on avalanche photodiodes (Supplementary Fig. 1a).

For functional mechanophenotyping of platelets based on molecular specificity in RT-FDC, platelets in PRP were labeled with a mouse anti-human monoclonal antibody CD61-PE (Beckman Coulter). Platelet activation was detected by direct immunofluorescence labeling of alpha granule release marker CD62P (P-selectin) with mouse anti-human monoclonal antibody CD62P-AlexaFluor647 (Clone AK4, Cat. No. 304918, BioLegend, USA)and activation associated conformational change in integrin αIIbβ3 is detected with a mouse anti-human monoclonal antibody PAC-1-FITC (Clone PAC-1, Cat. No. 340507, B.D. Biosciences, USA), respectively. PBS (Cat.No. P04-36500, PAN Biotech GmbH, Germany) and TRAP-6 (20 μM) (Haemochrom Diagnostica GmbH, Germany) were used as vehicle control and platelet agonist, respectively. Incubations were performed at room temperature for 10 min in the dark. Assessment of effects of actin disassembly by LatB and its effects on agonist-induced alterations in deformation in the presence of different ex vivo anticoagulants were performed as follows: for non-stimulated platelets, PRP was incubated (37 °C, in the dark) for 10 min with DMSO (0.99%) (Cat.No. D2650-5x5m, Sigma-Aldrich) or LatB (0.5 μg/ml) (cat.no. Cay10010631; Biomol GmbH). For measurement of TRAP-6 activated platelets, PRP was previously incubated (37 °C, in the dark) with DMSO (0.99 %) or LatB (0.5 μg/ml) for 10 min and then TRAP-6 (20 μM) (Haemochrom Diagnostica GmbH, Germany) was added for 10 min (37 °C, in the dark).

Deformability measurements were performed in a microfluidic chip with a constriction of 15 μm × 15 μm cross-section and a length of 300 μm (Flic15, Zellmechanik Dresden, Germany) (Supplementary Fig. 1b). Platelets in suspension are injected by a syringe pump (NemeSys, Cetoni GmbH, Germany), and cell deformation occurs due to the hydrodynamic pressure gradient created by the surrounding fluid only.[55]

Based on cellular circularity, deformation (Eq. 1) is calculated on the fly using bright-field images captured by a camera:[32]

$$\text{Deformation} = 1 - \frac{2\sqrt{\pi A}}{P} \qquad (1)$$

where $A$ is the cross-sectional area of the cell and $P$ its perimeter.

RT-FDC measurements were carried out in buffer CellCarrier B (Zellmechanik Dresden, Germany), which is composed of 0.6% (w/v) methylcellulose in PBS (without $Ca^{2+}$, $Mg^{2+}$). Here, 50 μL of immunofluorescently labeled PRP was suspended in 450 μL CellCarrier B. The PRP suspension was then driven through the microfluidic chip at flow rates of 0.006 μl/s, and the measurement was stopped after achieving 5000 single platelet count (hard-gate 150-33000 arbitrary units, A.U. for CD61-PE of fluorescence intensity) or after 10 min RT-FDC data was acquired using the ShapeIn software (Version 2.0, Zellmechanik Dresden, Germany). Using the Shape-Out analysis software (https://github.com/ZELLMECHANIK-DRESDEN/ShapeOut2/releases/tag/2.3.0) Version 2.3, Zellmechanik Dresden, Germany), kernel density estimation (KDE) plots of event density were generated, and statistical analysis was performed to determine the median values for platelet deformation, their size and the geometric mean of fluorescence (gMean) of the relevant functional variables. The range area ratio was limited to 0–1.1 and the cell size to 0–10 μm for the analysis (Supplementary Fig. 2 and 3).

**Flow cytometry**. Platelets were treated as described above for RT-FDC. We used PerFix-nc Kit (Cat.No. B31167; Beckman Coulter GmbH, Germany) and Phalloidin-Atto-647 (Atto-Tec GmbH, Germany) to measure changes in total F-actin content in the platelets. Flow cytometry data were processed using FlowJo™ software for Windows, Version v10.6.2. (Becton, Dickinson and Company, USA), and the gMean of the relevant variables was determined.

**Fluorescence microscopy**. Platelets in PRP were incubated with PBS (vehicle control, non-stimulated) or stimulated with TRAP-6 for 10 min followed by fixation in 2% paraformaldehyde (Morphisto, Germany) for 15 min Fixed platelets were transferred into a Shandon™ Single Cytofunnel™ (Thermo Fisher, USA) and were centrifuged on a microscope slide for 5 min at 700 rpm (Cytospin ROTOFIX 32 A, Hettich, Germany), washed thrice with PBS (5 min intervals). Platelet was permeabilized in 0.5 % saponin with 0.2% bovine serum albumin (BSA) (Cat. No. 11924.03, SERVA Electrophoresis GmbH, Germany) for 25 min, and followed by blocking for 30 min in 0.5% saponin supplemented with heat-inactivated 10% normal goat serum. Permeabilized platelets were incubated with 1:500 dilution of mouse monoclonal anti-α-Tubulin IgG (Clone DM1A, Cat. No. T9026, Sigma-Aldrich GmbH, Germany) primary antibody diluted in 0.5% saponin with 0.2% BSA in PBS for 16 h at 4 °C followed by three washing steps in PBS for 5 min each. Afterward, platelets were incubated with 1:750 dilution of goat polyclonal anti-mouse AF488 IgG prepared in 0.5% saponin with 0.2% BSA in PBS for 60 min in the dark at room temperature, followed by three washing steps with PBS for 5 min each. F-actin was labeled with 20 pM Phalloidin-Atto 647N (Cat. No. AD 647N-81, Atto-Tec GmbH, Germany) for 60 min, followed by three washes in PBS for 5 min each. Slides were covered by a permanent mounting medium Roti®-Mount Fluor-Care (Cat. No. HP19.1, Karl-Roth GmbH, Germany). Fluorescence microscopy was performed on a Leica SP5 confocal laser scanning microscope (Leica Microsystems, Wetzlar, Germany) equipped with HCX PL APO lambda blue 40×/1.25 OIL UV objective. For image acquisition, AF488 and ATTO647 were excited by argon (488 nm) and helium-neon (HeNe) (633 nm) laser lines selected with an acousto-optic tunable filter, and fluorescence emission was collected between 505–515 and 640–655 nm respectively on hybrid detectors. Assessment of F-actin distribution and organization of marginal band α-tubulin staining was performed by measuring cross-sectional line profile (5 μm length and 1 μm width) of non-saturated grayscale fluorescence intensities (pixel values) of immunofluorescent probes across individual platelets in confocal images using Leica Application Suite X (Version 3.7.1, Leica Microsystems, Wetzlar, Germany). For data plotting, GraphPad Prism version 8.0.0 for Windows (GraphPad Software, San Diego, California USA) was used.

**Statistical plots and analysis of RT-FDC data**. Statistical plots showing parameters of the platelet population were prepared with ShapeOut software (https://github.com/ZELLMECHANIK-DRESDEN/ShapeOut2/releases/tag/2.3.0 Version 2.3, Zellmechanik Dresden), PlotsOfDifferences (https://huygens.science.uva.nl/PlotsOfDifferences/), Raincloud Plots (https://gabrifc.shinyapps.io/raincloudplots/) and BoxPlotR (http://shiny.chemgrid.org/boxplotr/)[56,57].

**Statistics and reproducibility**. The number of single platelets analyzed, biological replicates, and the number of blood donors involved for each experiment are specified in the corresponding figure legend and figure description. All statistical assessments are performed in GraphPad Prism version 9.0.0 for Windows (GraphPad Software, San Diego, California USA). Details of specific statistical tests performed are described in figure descriptions, and $p < 0.05$ was considered significant.

**Reporting summary**. Further information on research design is available in the Nature Research Reporting Summary linked to this article.

## Data availability

Source data of corresponding figures are available in Supplementary Data 1. In addition, all raw data and accompanying software for analysis of RT-FDC data are available in a citable public repository and can be accessed directly at the following https://doi.org/10.5281/zenodo.4461273 or by requesting the corresponding author(s).

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

## Acknowledgements

O.O gratefully acknowledges funding from the German Ministry of Education and Research (BMBF) within the project 03Z22CN11 (ZIK grant) and the German Center for Cardiovascular Research within the project 81X3400107 (Postdoc start-up grant). This work was supported by the Deutsche Forschungsgemeinschaft project number 374031971–CRC/TR 240 project A06 to M.B., O.O. and R.P.

## Author contributions

L.S., A.G., M.B., O.O., and R.P. designed the study. L.S. performed all RT-FDC experiments. J.W. and L.L. performed flow cytometry. L.S. and R.P. performed CLSM experiments. L.S. and R.P. analyzed the data and prepared figures. L.S. wrote the manuscript. A.G. provided access to *MYH9* patient platelets. A.G., M.B., O.O., and R.P. contributed to writing the manuscript. All authors contributed to the critical revision of the manuscript. M.B., O.O., and R.P. obtained funding.

## Funding

## Competing interests

O.O. is co-founder and shareholder of Zellmechanik Dresden GmbH, distributing real-time deformability cytometry. All other authors declare no competing interests.
