## [Transparent Peer Review File · Communications Biology]

Reviewers' comments:

Reviewer #1 (Remarks to the Author):

The major claim of the paper is that some anticoagulants impact mechanical deformation and size of platelets. In particular, one of the tested anticoagulants, K2-EDTA, decreases both deformability and size of resting platelets, and abolishes the response to the agonist TRAP-6. The study provides important insights that can impact data interpretations involving mechanical properties of platelets. The following revisions are suggested.

1. It is possible that the effect of K2.EDTA on the observed phenotypes is simply due to using a higher concentration of K2.EDTA (1.8mg/ml, or ~4mM). Is it possible to identify a lower concentration of K2.EDTA where there is no longer the effect on mechanical properties of platelets and activation, but yet maintains the ability to prevent coagulation?
2. Mechanistically, EDTA is a Ca²⁺ chelator, and so one would expect platelets to be less contractile under K2.EDTA without agonist stimulation due to lack of calcium. However, EDTA decreases mechanical deformability (Fig. 1), and also upregulates F-actin compared to other anticoagulants without agonist stimulation (Fig. 5a). Thus, the results are somewhat surprising. A potential explanation needs to be provided in the main text.
3. It is well established that PAC-1 binding is both PH and Ca²⁺ sensitive, and thus it is not surprising that PAC-1 will not bind to EDTA-treated platelet. See: <https://www.bdbiosciences.com/eu/s/plateletactivation>
4. For CD62P expression, this previous paper reports somewhat opposite results: <https://pubmed.ncbi.nlm.nih.gov/22862794>. This discrepancy needs to be explained.
5. Figure 4 needs to be quantified with statistical analysis to support the claims in the main text.
6. Figure 5: If the claim is correct, does F-actin disruption by latrunculin impair TRAP-6 mediated decrease in deformability in different coagulants?
7. Figure 6: Statistical analysis is necessary to claim that the MYH9 mutation decreases deformability of platelets in ACD-A compared to healthy control, and what happens with TRAP-6 stimulation.

Reviewer #2 (Remarks to the Author):

In this manuscript, Sachs et al. examine the effects of different anticoagulants on the biomechanical properties of platelets using high-throughput real-time fluorescence and deformability cytometry (RT-FDC). They observe that K2-EDTA and Li-Heparin impair platelet deformation and decrease platelet size. K2-EDTA also blunted platelet activation following TRAP-6 stimulation as measured by CD62 surface expression and α IIB β 3 conformation change. The authors use confocal microscopy to evaluate the mechanism of action underlying these observations and conclude that EDTA-exposed platelets have increased baseline actin. They end the paper by showing that RT-FDC identifies the anticipated macrothrombocytopenia and altered deformability in platelets that carry an MYH9 mutation. However, the use of EDTA impairs MYH9-mutated platelet response to TRAP-6, preventing the investigators from accurately characterizing platelet behavior.

Sachs et al. argue that RT-FDC can be used to evaluate platelet function and assess bleeding risk in patients with platelet cytoskeletal defects. To that end, they assess the effect of different anticoagulants on measurements of platelet size and deformability to determine how to optimize this

assay. They make the novel observations that EDTA decreases baseline platelet deformability and use confocal data to support their hypothesis that this occurs due to inappropriate actin polymerization. These observations may help to lay a foundation for future studies using RT-FDC to characterize platelet function. The authors hint at but do not fully elucidate how EDTA may impact F-actin polymerization and platelet reactivity. I think it would increase the impact of the paper to try and better understand how EDTA and heparin are acting. Moreover, additional studies to characterize actin polymerization and configuration in the MYH9 platelets will improve our ability to interpret the RT-FDC results.

Major concerns:

- 1)The authors' observation that EDTA and to a lesser extent heparin, interfere with platelet size, deformability, and activation as quantified by RT-FDC are interesting. However, prior work has shown that EDTA and heparin interfere with platelet function and it would improve the quality and novelty of the paper to perform additional mechanistic studies to better understand the effect of EDTA on actin polymerization and platelet function.
- 2)The authors suggest that RT-FDC will be a useful technology to identify platelets with cytoskeletal defects and to assess future bleeding risk. However, their studies on MYH9 platelets are incomplete. It would be useful to repeat these studies with all the anticoagulants, including heparin, hirudin, and citrate – to strengthen their argument that ACD, citrate and hirudin are acceptable for use in this assay. Moreover, it would be useful to repeat the confocal studies in the MYH9 platelets to understand how decreased deformability on RT-FDC corresponds with actin configuration.
- 3)I recognize that this paper is focused on the effect of different anticoagulants on the results of RT-FDC, but I would be interested to see additional studies (or citations) to confirm that changes in platelet deformability measured by the assay correlate with platelet function. In other words, I think the paper would benefit from more of a discussion of the clinical utility of RT-FDC.

Minor concerns:

- 1)The color labeling of F-actin and α -tubulin in Figure 4b is reversed
- 2)In figure 4, it would be helpful to compare platelet diameter and actin fluorescent intensity in a bar-graph like those used in the previous figures. This would make it easier for the reader to understand the data and to display p values to demonstrate statistical significance.
- 3)In Figure 6 – I find the KDE scatter plots difficult to read and would appreciate the use of bar graphs as depicted in early figure for an easier comparison of changes in platelet size and deformability with the different anticoagulants before and after stimulation.

Reviewer #1 (Remarks to the Author):

The major claim of the paper is that some anti-coagulants impact mechanical deformation and size of platelets. In particular, one of the tested anticoagulants, K₂-EDTA, decreases both deformability and size of resting platelets, and abolishes the response to the agonist TRAP-6. The study provides important insights that can impact data interpretations involving mechanical properties of platelets.

Author response: We thank the reviewer for recognizing the relevance of *ex vivo* anticoagulants on biomechanical properties of blood platelets. Indeed, we also believe that our observations may be relevant for mechanical phenotyping of other peripheral blood cells

The following revisions are suggested.

1. It is possible that the effect of K₂.EDTA on the observed phenotypes is simply due to using a higher concentration of K₂.EDTA (1.8mg/ml, or ~4mM). Is it possible to identify a lower concentration of K₂.EDTA where there is no longer the effect on mechanical properties of platelets and activation, but yet maintains the ability to prevent coagulation?

Author response: The reviewer raises a relevant point here. Based on the reviewers suggestion we collected whole blood by phlebotomy in the presence of different concentrations (1.8mg/ml, 1.0mg/ml, 0.8mg/ml and 0.5mg/ml) of K₂-EDTA. We observed (see Figure 1 for reviewer, below) that coagulation occurred at the lowest K₂EDTA concentration (0.5mg/ml) and hemolysis occurred at K₂EDTA concentrations 0.8mg/ml and 1.0mg/ml. Based on these observations, the RT-FDC, flow cytometry, and imaging experiments of platelet-rich plasma (PRP) from the blood collected low K₂EDTA concentrations were not feasible.

Figure 1 (for reviewers only): Effect of different K₂-EDTA concentrations (1.8mg/ml, 1.0mg/ml, 0.8mg/ml and 0.5mg/ml) on human whole blood collected by phlebotomy.

2. Mechanistically, EDTA is a Ca²⁺ chelator, and so one would expect platelets to be less contractile under K₂.EDTA without agonist stimulation due to lack of calcium. However, EDTA decreases mechanical deformability (Fig. 1), and also

upregulates F-actin compared to other anticoagulants without agonist stimulation (Fig. 5a). Thus, the results are somewhat surprising. A potential explanation needs to be provided in the main text.

Author response: We agree with the 'reviewer's comments on the effect of calcium chelation. However, these results also took us by surprise. We have added the possible reasons behind this observation in the discussion part (please see highlighted part in revised version), which reads as follows:

Platelets collected in K₂-EDTA have the highest F-actin content under resting conditions in comparison to the other anticoagulants and show the lowest deformation. Our observations are in agreement with previous studies, which have demonstrated K₂-EDTA induced ultrastructural changes of the surface-bound canal system (narrowing and dilatation of the OCS) and an irreversible dissociation of the α Ib β 3 complexes^{43-45 46}. It is possible that the high content of F-actin in non-stimulated and TRAP-6 stimulated platelets and the associated platelet deformation could be explained by the irreversible dissociation of the α Ib β 3 complex and the associated cytoskeletal reorganization.

3. It is well established that PAC-1 binding is both PH and Ca²⁺ sensitive, and thus it is not surprising that PAC-1 will not bind to EDTA-treated platelet. See: <https://www.bdbiosciences.com/eu/s/plateletactivation>.

Author response: Indeed, the reviewer is correct, and we are aware of this, and we did take this critical information into account. Nevertheless, we decided to include mAbPAC-1 binding as an activation marker in the RT-FDC experiments to highlight that decreased biomechanical deformation of resting platelets in the presence of K₂-EDTA will not show PAC-1 binding compared to other *ex vivo* anticoagulants. Importantly, in addition to this, we used mAb PAC-1 binding to study the conformational changes of platelet integrin α Ib β 3 in Li-Heparin. By binding to α Ib β 3, Li-Heparin causes α Ib β 3-mediated outside-in signaling and induces cytoskeletal reorganization in non-stimulated platelets, which is associated with decreased deformation and increased F-actin content. Furthermore, since this is the first manuscript that comprehensively characterizes single platelet deformation mechanics and the associated platelet activation phenotypes simultaneously in RT-FDC, we believe these data are relevant for scientific documentation for other labs interested in RT-FDC and similar deformability cytometry-based approaches for mechanical characterization of blood platelets.

4. For CD62P expression, this previous paper reports somewhat opposite results: <https://pubmed.ncbi.nlm.nih.gov/22862794>. This discrepancy needs to be explained.

Author response: We thank the reviewer for bringing into discussion the paper entitled "Simultaneous assay of activated platelet count and platelet-activating capacity by P-selectin detection using K₂-EDTA-treated whole blood for antiplatelet agents" by Okano K et al.

Okano K et al. demonstrated that CD62P expression (% positive platelets) in platelet-rich plasma stimulated with 10 μ M ADP was higher in K₂EDTA than in sodium citrate. These different results could be justified due to the long incubation time of 20 min with the agonist and the constant stirring during incubation used by Okano et al., which is not the case in our experimental set-up (see methods section on RT-FDC). Schneider DJ et al. have shown that the effects of EDTA on platelet activation by agonists are related to the chelation of calcium and mainly vary as a function of time (PMID: 9386152 <https://pubmed.ncbi.nlm.nih.gov/9386152/>). In PRP, our data show that non-stimulated resting platelets in K₂EDTA have increased CD62P expression (measured as CD62P % positive platelets) compared with the other anticoagulants at the basal levels. TRAP-6 activated platelets show only a minor change in CD62P expression (% positive) in K₂EDTA compared with the other ex vivo anticoagulants. Furthermore, in a recent study, Aizawa et al., using flow cytometry, detected an decreased intracellular calcium pool and increased P-selectin expression in the presence of K₂EDTA.

Our results in whole blood showed (Figure 2 for reviewers, below) that TRAP-6-activated platelets in K₂EDTA showed CD62P expression (% positive) in whole blood but were lower than in ACD-A, Na-Citrate, Li-Heparin, and r-Hirudin. These results are comparable to the study by Okano K et al.

Figure 2 (for reviewers only): CD62P surface expression on platelets in whole blood in non-stimulated (resting) platelets and after TRAP-6 stimulation.

5. Figure 4 needs to be quantified with statistical analysis to support the claims in the main text.

Author response: We have now provided an updated graph of F-actin and alpha-tubulin, including statistical tests performed for both non-activated and TRAP-6 stimulated platelets in different *ex vivo* anticoagulants. Figure 4 and its description have been updated accordingly. We thank the

reviewer for suggesting this correction.

6. Figure 5: If the claim is correct, does F-actin disruption by latrunculin impair TRAP-6 mediated decrease in deformability in different coagulants?

Author response: This is a great question and relevant for gaining mechanistic insights into actin cytoskeleton-dependent platelet deformation and functional response.

We performed a new set of RT-FDC experiments to assess the effect of latrunculin-B (LatB) on F-actin-dependent alterations in platelet deformation. We used LatB at 0.5 $\mu\text{g/ml}$ and DMSO as vehicle control. In non-stimulated (resting platelets), our RT-FDC results show that platelets preincubated with LatB deform significantly more than vehicle control. Similarly, following activation with TRAP-6, LatB treated platelets deform significantly more than the TRAP-6 treated vehicle control (DMSO 0.99 %). (see Figure 3 for reviewer below).

Figure 3 (for reviewers only): Deformation of platelets in the presence of actin-depolymerizing agent LatB in different anticoagulants.

These experiments also show that the change in deformation (median) between vehicle control and LatB is greater for the anticoagulants K_2EDTA and Li-Heparin than for ACD-A and r-Hirudin. In addition, we assessed the influence of actin depolymerization by LatB on the surface expression of CD62P and integrin activation by PAC-1 binding. CD62P surface expression after TRAP-6 stimulation showed marked reduction upon LatB treatment in anticoagulants ACD-A, Li-Heparin, and r-Hirudin. In K_2EDTA , no change in CD62P expression occurred after TRAP-6 stimulation in the vehicle control and LatB, which confirms our data that CD62P expression is impaired after TRAP-6 activation in K_2EDTA . Our results are in agreement with the data from Woronowicz et al., which shows LatA (an isomer of latrunculin that binds gelsolin) inhibits the secretion of alpha-

granule by disrupting the actin cytoskeleton. We also observed a marked reduction in PAC-1 binding after LatB treatment in unstimulated platelets with the anticoagulants ACD-A, Li-Heparin, and r-Hirudin. Interestingly, upon TRAP-6 activation, a significant reduction in PAC-1 binding is evident only with Li-heparin. Taken together, these data suggest that by blocking actin polymerization and destabilizing the actin cytoskeleton with LatB, the marked difference in deformation in K₂EDTA mainly arises due to the increased F-actin content in non-stimulated platelets.

These data are now presented in the new results section as Figure 6 in the revised main text and Supplementary Figure 8 in supporting information. Again, we thank the reviewer for suggesting this insightful set of experiments.

7. Figure 6: Statistical analysis is necessary to claim that the MYH9 mutation decreases deformability of platelets in ACD-A compared to healthy control, and what happens with TRAP-6 stimulation.

Author response: We thank the reviewer for the suggestion. New figure sub-panels with statistical graphs in Figure 6, including the significance values derived from the statistical analysis, have been added in the revised version.

Reviewer #2 (Remarks to the Author):

In this manuscript, Sachs et al. examine the effects of different anticoagulants on the biomechanical properties of platelets using high-throughput real-time fluorescence and deformability cytometry (RT-FDC). They observe that K₂-EDTA and Li-Heparin impair platelet deformation and decrease platelet size. K₂-EDTA also blunted platelet activation following TRAP-6 stimulation as measured by CD62 surface expression and α IIb β 3 conformation change. The authors use confocal microscopy to evaluate the mechanism of action underlying these observations and conclude that EDTA-exposed platelets have increased baseline actin. They end the paper by showing that RT-FDC identifies the anticipated macrothrombocytopenia and altered deformability in platelets that carry an MYH9 mutation. However, the use of EDTA impairs MYH9-mutated platelet response to TRAP-6, preventing the investigators from accurately characterizing platelet behavior.

Sachs et al. argue that RT-FDC can be used to evaluate platelet function and assess bleeding risk in patients with platelet cytoskeletal defects. To that end, they assess the effect of different anticoagulants on measurements of platelet size and deformability to determine how to optimize this assay. They make the novel observations that EDTA decreases baseline platelet deformability and use confocal data to support their hypothesis that this occurs due to inappropriate actin polymerization. These observations may help to lay a foundation for future studies using RT-FDC to characterize platelet function. The authors hint at but do not fully elucidate how EDTA may impact F-actin polymerization and platelet reactivity. I think it would increase the impact of the paper to try and better understand how EDTA and heparin are acting. Moreover, additional studies to characterize actin polymerization and configuration in the MYH9 platelets will improve our ability to interpret the RT-FDC results.

Author response: We thank the reviewer for this excellent and concise summary of our manuscript. As remarked by the reviewer, our primary goal in the current manuscript is to provide an in-depth analysis of single platelet biomechanics by high-throughput functional mechanophenotyping in RT-FDC. The reviewer also highlights some key aspects of our study, which need further characterization, particularly the observations related to the effect of K₂-EDTA and Li-Heparin. In the following responses to the reviewer's queries, we provide additional data and incorporate all changes corrections suggested by the reviewer.

Major concerns:

1) The authors' observation that EDTA and to a lesser extent heparin, interfere with platelet size, deformability, and activation as quantified by RT-FDC are interesting. However, prior work has shown that EDTA and heparin interfere with platelet function and it would improve the quality and novelty of the paper to perform additional mechanistic studies to better understand the effect of EDTA on actin polymerization and platelet function.

Author response: We thank the referee for the invaluable suggestion. Indeed, this was also recommended by referee 1 (see above). Based on this, we assessed platelet deformation and their functional response in the presence of Latrunculin B in different *ex vivo* anticoagulants. In the revised version, we have added a new results section, updated methods, and new data panels (Figure 6 and Supplementary Figure 8) that give mechanistic insights into actin polymerization status and its impact on platelet deformation and functional response. In brief, platelets preincubated with latrunculin B become significantly more deformable than in the vehicle control. Similarly, following activation with TRAP-6, LatB treated platelets are significantly more deformable than the TRAP-6 treated vehicle control (DMSO 0.99 %). (see rebuttal Figure 3 above).

2)The authors suggest that RT-FDC will be a useful technology to identify platelets with cytoskeletal defects and to assess future bleeding risk. However, their studies on MYH9 platelets are incomplete. It would be useful to repeat these studies with all the anticoagulants, including heparin, hirudin, and citrate – to strengthen their argument that ACD, citrate and hirudin are acceptable for use in this assay.

Moreover, it would be useful to repeat the confocal studies in the MYH9 platelets to understand how decreased deformability on RT-FDC corresponds with actin configuration.

Author response: We agree with the reviewer it will be desirable to assess the deformability experiments in other anticoagulants. Soon after receiving the peer review assessment of our manuscript, we discussed this matter in detail with the consulting hematologist (co-author Prof. Dr. Andreas Greinacher MD). It was brought to our notice that the MYH9 patient is a mother of three young children and may not be able to travel to Greifswald on short notice. In addition, due to current COVID-19 travel restrictions and safety measures mandated by the local health authorities, it is not advised to schedule patient appointments at our hospital research lab solely for the purpose of research investigations. Furthermore, on a critical patient safety issue related to blood donation from individuals with MYH9 bleeding diseases, we cannot withdraw several vacutainers with different *ex vivo* anticoagulants, each with > 8mL of blood at a single sitting, which is a limiting factor. Thus in light of the above circumstances, at this moment, we may not be able to provide additional new experiments on MYH9 patient platelets beyond what is already comprehensively characterized in the presence of *ex vivo* anticoagulants ACD-A and EDTA. This information was promptly communicated with Dr. Alexander Cartagena-Rivera, Editorial Board Member, and with Dr. Anam Akhtar, Associate Editor at Communications Biology, on the 5th of July 2021.

We anticipate that the situation may change in due course as COVID-19 vaccination rates increase and travel restrictions ease.

Confocal imaging of MYH9 platelets to understand how decreased deformability on RT-FDC corresponds with actin configuration.

Author response: We thank the reviewer for this suggestion. In collaboration with Dr. Markus Bender (University Hospital Wuerzburg, Germany), we are preparing a manuscript that precisely investigates this aspect and covers some of the most common MYH9 mutations, such as the D1424N, E1841K and R702C observed in patients. In these studies, we are using relevant murine models and patient samples to comprehensively assess the influence of these mutations on the platelet cytoskeleton using cell-biological and biophysical approaches. We hope to present these results soon to the broader scientific community.

3) I recognize that this paper is focused on the effect of different anticoagulants on the results of RT-FDC, but I would be interested to see additional studies (or citations) to confirm that changes in platelet deformability measured by the assay correlate with platelet function. In other words, I think the paper would benefit from more of a discussion of the clinical utility of RT-FDC.

Author response: We thank the reviewer for this suggestion. So far using RT-FDC this is the first manuscript that comprehensively assesses simultaneously the relationship between platelet deformation and their corresponding functional state. In the revised version of the manuscript we have added a separate paragraph on clinical and translational utility of RT-FDC with relevant references.

The text is as follows:

Clinically relevant translational applications of real-time deformability cytometry for the morpho-rheological and biomechanical characterization of a variety of cells are rapidly emerging. Clinically relevant translational applications of RT-DC in the diagnosis of disease states by morpho-rheological and biomechanical characterization of a variety of cells by RT-DC are fast emerging. These include biomechanical differentiation for diagnosis of hereditary spherocytosis and malaria infections in peripheral blood cells ⁴¹ and in biophysical fingerprinting of primary human skeletal stem cells from bone marrow ⁴². In addition, the feasibility of RT-DC in routine quality control of platelet concentrates and transplantable hematopoietic stem cells in blood banks is currently being tested ⁴³. More recently Kubankova, M., et al. using RT-DC performed biomechanical fingerprinting of erythrocytes, lymphocytes, monocytes, neutrophils, and eosinophils in clinical induced by severe acute respiratory syndrome corona virus2 (COVID-19 disease) ⁴⁴. The study results showed significantly more deformable lymphocytes and neutrophils, increased size of monocytes, lymphocytes, and neutrophils, and the appearance of smaller and less deformable erythrocytes.

Minor concerns:

1)The color labeling of F-actin and α -tubulin in Figure 4b is reversed

Author response: The color mislabeling has been corrected in the revised version.

2) In figure 4, it would be helpful to compare platelet diameter and actin fluorescent intensity in a bar-graph like those used in the previous figures. This would make it easier for the reader to understand the data and to display p values to demonstrate statistical significance.

Author response: Figure 4 has been accordingly updated in the revised version of the manuscript. We thank the reviewer for the excellent suggestion.

3)In Figure 6 – I find the KDE scatter plots difficult to read and would appreciate the use of bar graphs as depicted in early figure for an easier comparison of changes in platelet size and deformability with the different anticoagulants before and after stimulation.

Author response: We understand the reviewer's concern with respect to the readability of KDE plots from RT-DC data. Therefore, based on reviewer's suggestion, Figure 6 (now Figure 7 in revised version) has been updated with statistical rain cloud plots that separately represent platelet deformation and size before and after stimulation in different *ex vivo* anticoagulants derived from the KDE plots.

REVIEWERS' COMMENTS:

Reviewer #1 (Remarks to the Author):

The authors have satisfactorily addressed all of the comments.

Reviewer #2 (Remarks to the Author):

I appreciate the responsiveness of the authors to the reviewers suggestions. The new studies they performed using LatB to demonstrate that the decrease in platelet deformability seen with EDTA is due to increased actin polymerization. Although the effect of EDTA on integrin activation is not novel, this study confirms that it should not be used in platelet function assays.

I also understand that the challenge of obtaining more clinical samples, particularly from pediatric patients, and of course agree it is not necessary to repeat the MYH9 studies with all anticoagulants. However, 1 sentence describing the authors reasoning behind using ACD as the comparator to EDTA would be helpful, particularly when sodium citrate is the anticoagulant most commonly used for platelet aggregation studies performed clinically.

Overall, this study is a rigorous investigation of use of different anticoagulations in RT-FDC to help optimize this technology for future research and clinical use that also sheds additional light on the effect of different anticoagulants on platelet reactivity.

Responses to the reviewers

Reviewer #1 (Remarks to the Author):

The authors have satisfactorily addressed all of the comments.

Author response: Thank you.

Reviewer #2 (Remarks to the Author):

I appreciate the responsiveness of the authors to the reviewers suggestions. The new studies they performed using LatB to demonstrate that the decrease in platelet deformability seen with EDTA is due to increased actin polymerization. Although the effect of EDTA on integrin activation is not novel, this study confirms that it should not be used in platelet function assays.

I also understand that the challenge of obtaining more clinical samples, particularly from pediatric patients, and of course agree it is not necessary to repeat the MYH9 studies with all anticoagulants. However, 1 sentence describing the authors reasoning behind using ACD as the comparator to EDTA would be helpful, particularly when sodium citrate is the anticoagulant most commonly used for platelet aggregation studies performed clinically.

Overall, this study is a rigorous investigation of use of different anticoagulations in RT-FDC to help optimize this technology for future research and clinical use that also sheds additional light on the effect of different anticoagulants on platelet reactivity.

Author response: We thank the reviewer for their interest in our study and constructive feedback. We have added the following sentence for choosing ACD-A over Na-Citrate for the MYH-9 patient platelet analysis.

“Concerning our reasoning behind choosing ACD-A over Na-Citrate as an ex vivo anticoagulant as a comparator to K2-EDTA for the analysis of platelets from a patient with MYH9 mutation in RT-FDC, there are several lines of experimental evidence which indicate ACD-A is superior to Na-Citrate in terms of maintaining platelet physiology. In practice, blood collection systems anticoagulated with Na-Citrate are primarily used for studies of plasmatic coagulation and aggregation studies. Intriguingly, Na-Citrate may induce the formation of micro-aggregates, thus leading to a decrease in platelet count over time (PMID: 19172515 and PMID: 21999185). On the contrary, ACD-A is a more physiological anticoagulant capable of maintaining platelet physiology and signal transduction mechanisms with minimal impact on platelet responsiveness to agonists (PMID: 7714666).”